# Recent advances in cryotolerance biomarkers for semen preservation in frozen form–A systematic review

Jiajia Suo[1,2], Jieru Wang[1], Yanling Zheng[2], Fayi Xiao[2], Ruchun Li[2], Fei Huang[1], Peng Niu[1], Wei Zhu[2], Xiaoxia Du[2], Jianxiu He[2], Qinghua Gao[1]*, Ahrar Khan[2,3]*

1 College of Life Sciences, Tarim University, Alar, Xinjiang, China, 2 Department of Veterinary Medical Science, Shandong Vocational Animal Science and Veterinary College, Weifang, Shandong, China, 3 Faculty of Veterinary Science, University of Agriculture, Faisalabad, Pakistan

* gqhdky@126.com (QHG); ahrar1122@yahoo.com (AK)

**Data Availability Statement:** All data are in the manuscript and supporting information files.

**Funding:** Research on Key Technologies supported this study for Improving the Quality of Beef Cattle

## Abstract

Spermatozoa cryopreservation has been practiced for decades and is a very useful technique for long-term preservation of sperm fertility. The capability for semen cryopreservation varies across species, seasons, latitudes, and even for different ejaculates from the same animal. This article summarizes research results on sperm cryotolerance biomarkers in several species, focusing on three areas: spermatozoa cryotolerance biomarkers, seminal plasma proteins cryotolerance biomarkers, and other cryotolerance biomarkers. We discovered that sperm cryoresistance biomarkers are primarily related to sperm plasma membrane stability, the presence of antioxidant substances in sperm or seminal plasma, sperm cell energy metabolism, water and small molecule transport channels in the sperm plasma membrane, and antistress substances in sperm or seminal plasma. The research conducted using diverse livestock models can be employed to enhance the basic and applied reproduction of other mammals through the study of sperm cryotolerance biomarkers, as well as the substantial similarities between livestock and other organisms, including endangered species.

## Introduction

Spermatozoa cryopreservation has been used for decades and is the useful method of preserving male fertility for long time, while assisted reproductive procedures, genetic enhancement, and biobanking are some of the clinical and research applications of this approach. Sperm cryopreservation has been used in livestock for breeding program management, genetic enhancement, infertility control and genomic resourcefulness banking for endangered species [1]. Sperm cryopreservation has been attempted in boars, bulls, camels, goats, rams, and other species [2] since 1940s [3, 4]. In practice, sperm cryopreservation enables their enduring storage, propagation of genetically superior animals in future generations, and transportation of semen to far flange distances [5, 6]. Another superiority of employing frozen-thawed sperm is that in spite of relying on the physical existence of the breeding bull on the livestock farm, animal growers instantly impregnate dairy cows at the appropriate time. Cryopreservation is also

Population in Southern Xinjiang (2021BB011), Tarim University Graduate Research Innovation Program Funded Project (TDBSCX202108), the Shandong Provincial Modern Agricultural Industry Technology System (SDAIT-09-07), Shandong Province Collaborative Promotion Plan of Agricultural Major Technology in Shandong Province (SDNYXTTG-2023-07), and The Shandong Province Science and Technology SMEs Innovation Capability Improvement Project (2022TSGC2325, 2023TSGC0714). This work was also supported by Traim University (Xinjiang, China) and Shandong Vocational Animal Science and Veterinary College (Shandong, China).

**Competing interests:** The authors have declared that no competing interests exist.

an important process for the administration of germplasm banks, which supports biodiversity preservation and safeguard of endangered species [7]. Despite its wide utility, cryopreserved spermatozoa can lose physiologic characteristics such as viability and motility, even when optimized procedures are applied [8].

According to freezing protocol, there are differences in the ability to withstand frozen-thawed processes among ejaculates in almost all species. Additionally, not all semen is suitable for freezing, and not all semen samples with good viability freeze well, and there is significant variability between and within ejaculates, even between fractions of the same ejaculate. Ejaculates have been categorized as either good freezability ejaculates (GFE) or poor freezability ejaculates (PFE) based on the results of cryopreservation [9]. While having greater genetic value and satisfactory reproductive success with natural mating or use of fresh semen for artificial insemination, a group of ejaculates can show little to no sperm cryotolerance [2]. As a result, use of the conventional index to forecast the cryoresistance of fresh semen is inaccurate. Nonetheless, some researchers have focused on predicting sperm cryotolerance in agriculturally important species. The post-thaw motility of frozen-thawed ejaculates acceptable for insemination is measured using computer-assisted semen analysis (CASA), and further tests have been established to predict the fertility of male sperm more accurately. These tests include the assessment of mitochondrial activity, DNA and acrosome integrity and hypo-osmotic resistance [10]. Furthermore, research efforts are now focused on identifying trustworthy fertility biomarkers in the sperm and seminal plasma, and other related factors [11]. The main biomarkers of spermatozoa cryotolerance reported so far are given in Table 1.

The research of sperm freezing tolerance biomarkers can forecast the reproductive ability of sperm post-thawing, considerably decreasing the wastage of valuable time, personnel efforts, material, and financial resources. The study on the molecular mechanism of sperm freezing tolerance can provide theoretical support for sperm cryopreservation, as well as give a theoretical foundation and optimization direction for the semen freezing techniques. All such studies can help increase breeding income, enhance frozen sperm production efficiency, and promote

**Table 1. Cryotolerance biomarkers in sperm.**

| Biomarker | Species | Function during freeze/thaw protocol | References |
|---|---|---|---|
| GST | Bull | Cell protection against oxidative stress | [5] |
| GSTM3 | Boar | Cell protection against oxidative stress | [9] |
| Cu/ZnSOD | Boar | Antioxidant machinery | [13] |
| CFTR | Bull | The activity of antioxidant enzymes and sperm mitochondrial function | [16] |
| GLUT3 | Boar | Supplies energy from glucose substrates for protein phosphorylation | [13] |
| HSP90AA1 | Boar | Thermal shock, apoptosis, scavenging for ROS, and sperm motility | [13] |
| NT5C1B | Bull | Regulation of nucleotide metabolism | [1] |
| FH | Bull | The enzyme of the TCA cycle | [1] |
| proAKAP4 and AKAP4 | Stallion, Bull | Flagellum structuration and motion, sperm fibrous skeleton protein, regulation of $Ca^{2+}$ influx and motility | [17, 66] |
| Tyrosine Phosphorylation State | Bull | Participate in acrosome reaction, | [18] |
| DHA, stearic acid, and PUFA | Stallion | Give the membrane greater fluidity and flexibility | [22] |
| HSP70 | Boar | Anti-stress molecular mechanisms | [23] |
| AQP3 and AQP7 | Bull, Boar Stallion | Safeguarding against osmotic alterations by effectively managing the flow of glycerol and water in the sperm membrane. These biomarkers also required in sperm motility. | [28–34] |
| AQP11 | Bull, Boar, Stallion | Safeguarding against osmotic alterations by effectively managing the flow of glycerol and water in the sperm membrane. | [32, 33] |

the use and development of efficient reproduction technology. This review focuses on male sperm cryotolerance and aims to address all of related biomarkers.

## Methods

### Search of literature

Publications were detected screening at first level in PubMed and Google scholar database till the end of March 2023. The following terms were used to search for the widest field possible: "marker" and "sperm cryopreservation", "biomarker" and "sperm cryopreservation", "marker" and "sperm cryotolerance", "marker" and "sperm cryoresistance". All articles searched were manually assessed to alleviate the risk of overlooking any further possible related articles.

After conducting the search in the databases, a standardized data extraction was performed to obtain the articles listed in PMID format, with each entry separated by tabs. The extracted information included the PubMed ID, journal, ISSN, document type, publication date, authors, title, DOI, keywords, and abstract. Subsequently, an Excel sheet was generated to include all of this information. To determine appropriateness, all data in the Excel file was independently evaluated by two researchers and then re-examined for any discrepancies one co-author. Articles were excluded if not written in English or did not report indicators of sperm cryotolerance or sperm freezing resistance. Publications were considered if they reported indicators of sperm cryotolerance or sperm freezing resistance. Finally, the full text of chosen articles was downloaded and carefully read, and the contents were analyzed. This all process led to a final list of articles on which this systematic review is based.

### Article quality assessment

The Newcastle-Ottawa scale (NOS) was employed to evaluate the appropriateness of Publications. The NOS assesses selection, comparability, and outcome. It has three versions for the evaluation of case-control, cross-sectional studies, and cohort studies, specifying 7 and 8, respectively. Although a minor difference was noted in their great quantity, each one is valued with an asterisk, except for the "comparability" criterion; the maximum score possible is 9. We must highlight here that the works that assembled less than 5 points are considered possibly biased. Remarkably, there weren't any studies excluded under the quality criterion.

Once studies were chosen, those were evaluated for the following data: 1) aim of the study, 2) species, 3) markers, 4) methods, 5) molecular mechanisms, 6) sperm functional quality/fertility parameters, 7) main results, 8) conclusions, and 9) references. An additional row inserted indicating the types of markers for sperm cryotolerance (Biomarkers from spermatozoa or seminal plasma, or others).

A total of 336 articles during the search process were retrieved. Subsequently, redundant articles were eliminated, and we had 152 articles. Meanwhile, the articles were screened in three independent steps and ultimately, this systematic review is comprised of 26 studies.

## Results

Tables 1–3 summarized the study characteristics about various cryotolerance biomarkers of spermatozoa, and seminal plasma. The PRISMA 2020 flowchart was utilized for presentation of selection process of the studies for systematic review (Fig 1).

### Biomarkers of spermatozoa

**Antioxidant protection.** The objective of this review was to look for biomarkers of a male ejaculates of potentially "Good" or "Poor" ability to sustain cryopreservation by assessing the

**Table 2. Cryotolerance biomarkers of seminal plasma.**

| Biomarkers | Species | Function during freeze/thaw protocol | References |
|---|---|---|---|
| TAC | Stallion | total antioxidant capacity | [22] |
| IGF1 | Bull | play a cryoprotective role in antioxidative protection during semen freezing | [39] |
| HSP2 | Stallion | Protection for sperm membrane properties during semen freezing procedures | [40] |
| β-HEX | Boar | Lead to changes in plasma membrane liquidity and improve sperm mitochondrial function | [42] |
| NPC2 | Boar | Bind the plasma membrane cholesterol | [44] |
| L-PGDS | Bull Boar | Participate in modifications of cell membrane permeability. | [38, 44] |
| HSP90a | Boar | Participates in sperm capacitation and apoptosis processes, and membrane integrity | [44] |
| BSP A1/A2 | Bull | Protect the sperm membranes | [38] |
| aSFP | Bull | Protect sperm membrane against peroxidation. | [38] |
| NGF | Bull | Protecting sperm membrane integrity against cryoinjury | [50] |
| EBU2P3 | Bull | Protecting mitochondrial membrane potential during cryopreservation | [50] |
| CRISP-3 | Stallion | Overlapping in reproduction, | [51] |

proteins involved in sperm cell physiology. There have been few studies on biomarkers of sperm freezing resistance for a long time, with most studies focusing on proteins or biological factors of spermatozoa. Llavanera et al. [9] discovered that variations and deficits of sperm-oocyte binding proteins at the time of fertilization are one of the main reasons of failure in reproduction at *in vitro* as well as *in vivo* and that one of the sperm-oocyte binding proteins such as glutathione S-transferase Mu 3 (GSTM3) may possibly be used as a cryoresistance marker of spermatozoa in boars. While Ryu et al. [5] stated that the Glutathione S-transferases (GSTs) are multi-gene isoenzymes that protect cells against oxidative stress caused by both xenobiotic and endobiotic substances in bulls. Furthermore, distinct isoforms of GSTs have varying levels of expression in different tissues, and the two isoforms of GSTs are strongly linked to cryotolerance in ejaculated spermatozoa and epididymal spermatozoa [12].

It has also been indicated that the copper- and zinc-containing superoxide dismutase (Cu/ZnSOD) is tangled in the antioxidant activity, as it catalyzes the dismutation of superoxide anions into $H_2O_2$ and $O_2$, and it has been proposed that these performance an significant role in boar sperm endurance after cryopreservation [13]. It is also known that SOD contents were not only determined in spermatozoa but also in the seminal plasma in jackass [14].

To date, the variations in sperm competence to endure freeze-thawing protocols have mostly been reported on the basis of post-thaw sperm motility and membrane integrity, though Yeste et al. [15] suggested that reactive oxygen species (ROS) production and mitochondrial membrane potential are also linked to cryoresistance in the stallion spermatozoa. According to a study of candidate genes for sperm motility [16], a missense mutation within

**Table 3. Other cryotolerance biomarkers of spermatozoa.**

| Biomarkers | Species | Function during freeze/thaw protocol | References |
|---|---|---|---|
| Apoptotic markers | Stallion | Sperm apoptosis | [54] |
| Calcium Ionophore Induced Membrane Changes | Dog | Lead to at least a part of the acrosomal deterioration | [55] |
| Small extracellular vesicles miRNAs | Boar | Concerned in low sperm cryotolerance probably detrimental modulation of essential pathways concerned to energy production and sperm development | [57] |
| Fe | Bull | Associated with oxidative damage to DNA, lipids, proteins. Fe also influences DNA integrity and tail membrane | [59] |
| Se | Bull | Associated in spermatogenesis and is an fundamental part of GPX4 | [59] |
| Zn | Bull | Concerned in maturation of epididymal sperm and capacitation | [59] |

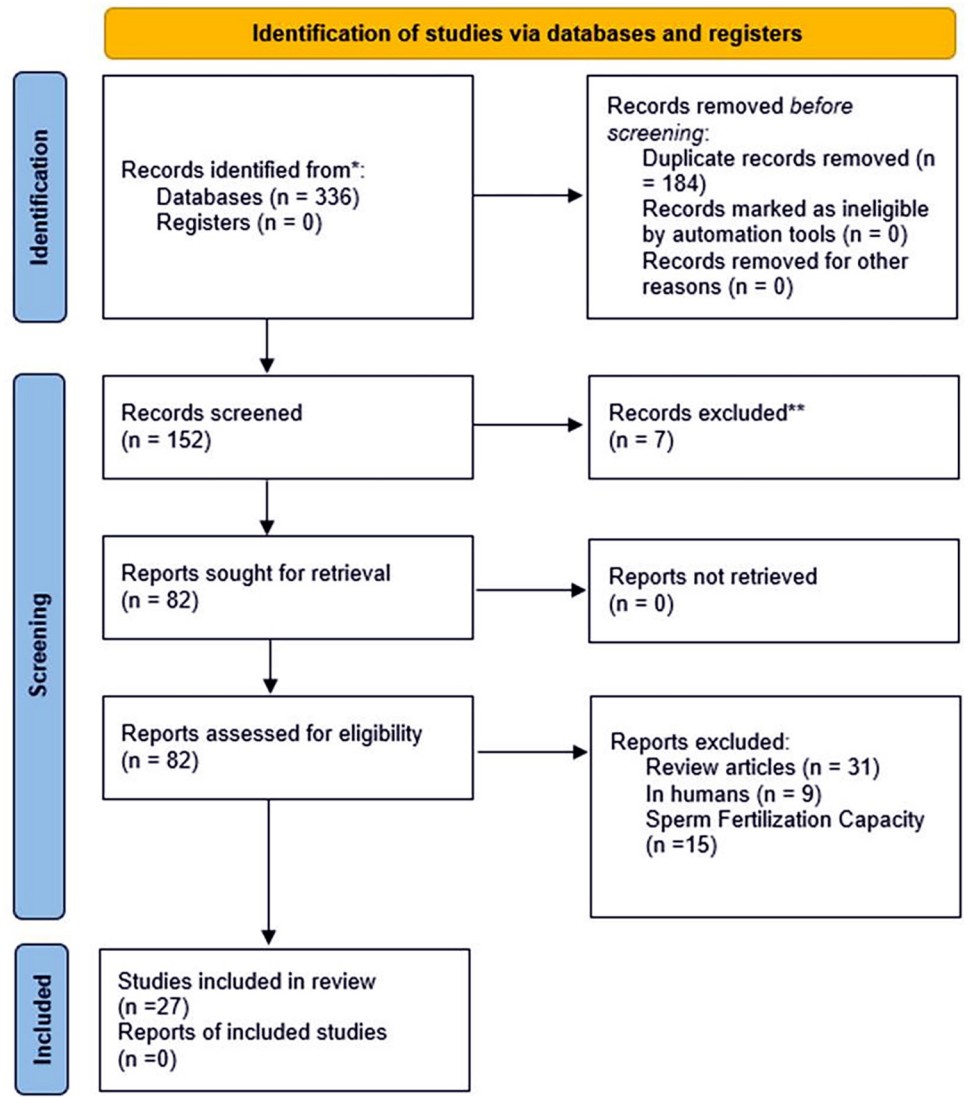

**Fig 1. PRISMA 2020 Flow diagram of the study selection for systematic review.** Studies reporting association: 15 studies in this review reported 18 biomarkers of spermatozoa, 9 studies in this review reported 13 biomarkers of seminal plasma, 4 studies in this review reported 6 other cryotolerance biomarkers.

exon 11 of the cystic fibrosis trans-membrane conductance regulator (CFTR) gene (Met468Leu) is associated with antioxidant enzyme activity and mitochondrial function of frozen-thawed sperm in Holstein-Friesian bulls. All the biomarkers present in sperm are related to ROS production, oxidative stress, or antioxidant enzyme activity which may play a critical role in scavenging ROS formed from the reduction of oxygen during cryopreservation. According to Casas et al. [13], the glucose transporter 3 (GLUT3), the heat shock protein 90-kDa alpha A1 (HSP90AA1), and the Cu/ZnSOD play critical roles in the sperm survival post cryopreservation, while GLUT3 provides energy from glucose substrates for protein phosphorylation and affects the input of energy to the boar sperm.

**Sperm metabolism.** In an investigation of functional and proteomic differences between bulls of GFE and PFE type, Song et al. [1] found that anti-cytosolic 5-nucleotidase 1B (NT5C1B) and anti-fumarate hydratase (FH) are closely associated with cryoprotectant

tolerance in bulls, for that NT5C1B is allied with the adjustment of nucleotide usage while FH is associated with the Krebs/tricarboxylic acid/citric acid cycle. While other scientists suggest that A-kinase anchor protein 4 (AKAP4) and proAKAP4 concentration is a promising biomarker of sperm quality and thawed spermatozoa quality of stallion [17], the flagellar shape, chemotaxis, capacitation, and sperm motility are all influenced by AKAP4. In addition, the flagellum structure and motion, sperm fibrous skeleton protein, regulation of $Ca^{2+}$ influx and motility are influenced as well. According to a study explaining the link between protein tyrosine phosphorylation state and sperm characteristics [18], immunodetection levels of tyrosine-phosphorylated proteins are valid markers that can predict the tolerance to frozen storage in spermatozoa of Japanese black bulls.

It appears that some of the biomarkers are associated with sperm metabolism, like GLUT3 regulates protein phosphorylation and energy input [13], NT5C1B affects nucleotide metabolism [1], while FH affects the tricarboxylic acid cycle, AKAP4 and proAKAP4 control a few sperm metabolisms [17], and the protein tyrosine phosphorylation state of sperm which could also be used as a marker. As a result, the metabolism of three major nutrients is influenced in spermatozoa, and this alteration has an impact on the plasma membrane, cytoskeleton, motor metabolism, and energy supply of sperm during the freezing procedure, and the key factors involved in sperm nutrient metabolism may be biomarkers of sperm freezing resistance in future studies.

While extensive knowledge exists regarding fatty acids, and their role as energy sources and structural components within cells, their involvement in fertility and cryopreservation remains less understood [19]. In studies on separation and analysis of the key fatty acids (Fig 2) from stallions classed as GFE and PFE, lauric, myristic, and oleic acids were found to be nearly 2-fold more plentiful in the sperm cells of the GFE compared to the PFE, which could be used as indicators for GFE [20]. Moreover, sperm containing significant quantities of stearic acid and docosahexaenoic acid (DHA) (18:0) demonstrates superior post-cryopreservation motility compared to those with lower levels. This phenomenon extends to sperm rich in polyunsaturated fatty acids (PUFA). However, the exact mechanism inherent to functionality of fatty acid

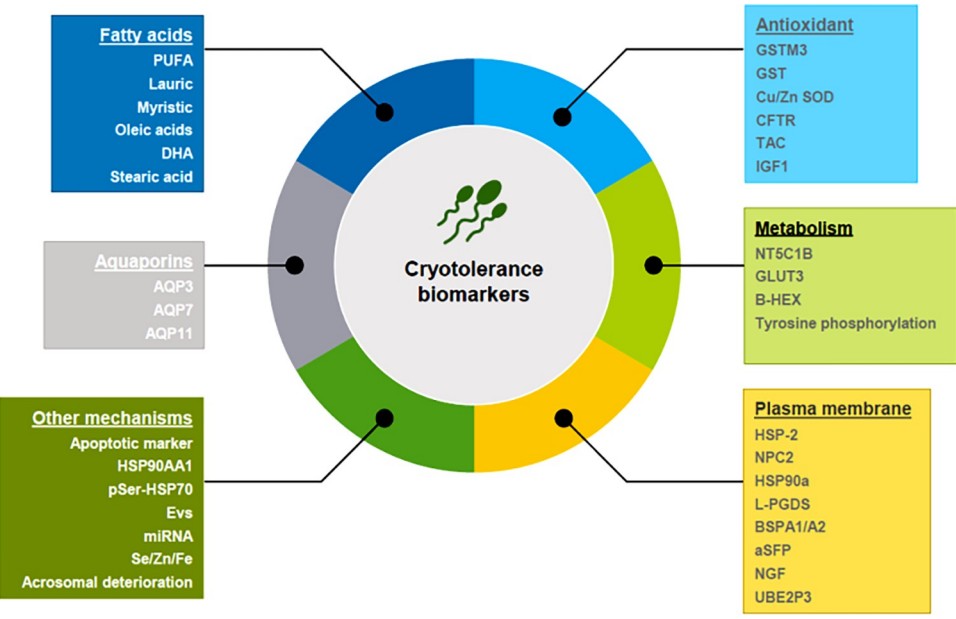

**Fig 2. Types of cryotolerance biomarkers of spermatozoa.**

in cryotolerance remains unclear, even though cells with higher levels of PUFA have an enhanced level of membrane flexibility and fluidity that is granted by PUFA with many double bonds [21, 22].

**Molecular chaperone.** The pSer levels of HSP70 can predict sperm cryoresistance in the research of the parameters examined in spermatozoa to delve deeper into the mechanisms required to enhance the effects of heat shock on boar-sperm cryotolerance at 17˚C [23]. While the HSP90AA1 belongs to the same chaperone protein family as HSP70 and is involved in heat shock, inhibits apoptosis, scavenging for ROS, and sperm motility, while all these functions determine the phosphorylation of proteins in the process of capacitation [13]. Many chaperone proteins are classified as stress proteins, with one important function in preventing the polymerization of newly synthesized polypeptide chains into non-functional structures, to prevent the incorrect combination between non-natural states, to increase the correct folding of proteins, and slow down the denaturation of proteins during the procedure of semen freezing and thawing.

Aquaporin (AQP) stands out as one of the most extensively studied and comprehensive indicators of sperm freezing resistance (Fig 2). Discovered by Agre [24] in 1992, his investigations into the structural and functional characteristics of AQPs, along with their tissue distribution, culminated in the Nobel Prize in Chemistry in 2003. Certain AQPs also exhibit permeability to minor solutes such as glycerin [25, 26], comprising a class of widely distributed transmembrane proteins facilitating enhanced water passage. Notably, the pivotal aspect of the sperm cryopreservation process involves the transit transfer of small molecular solutes and water between sperm and the cryopreservation solution, underscoring the crucial role of AQPs as biomarkers for sperm cryopreservation. A total of 13 AQPs (AQP0-AQP12) have been found in mammalian cells so far, and they have been divided into three groups: orthodox AQPs, aquaglyceroporins (GLPs), and superAQPs [27, 28]. According to research that used various inhibitors to reveal the functional importance of each group of AQPs during sperm cryopreservation, GLPs played a critical role during stallion sperm cryopreservation [29].

AQP3, AQP7, and AQP11 are related with cryoresistance, cryopreservation, and fertilizing capacity of cryopreserved spermatozoa from bull [30–32], boar [28, 33], and stallion [29, 34]. Localization of AQP3 (bull and boar), AQP7 (bull and stallion), and AQP11 (bull, boar, and stallion) has been reported in principal piece of the sperm tail while in mid-piece, AQP3 (bull, boar, and stallion), AQP7 (bull and boar), and AQP11 (bull and boar) have been reported. In the neck region of spermatozoa, so far AQP7 has been found in boar. In the head region, AQP3 (bull and boar), and AQP7 and AQP11 (bull, boar, and stallion) has been reported [28–34]. The association between AQP3, AQP7 and AQP11 and sperm cryotolerance differs between boar and bull. The AQP7 is connected to sperm resilience to endure post-thawing in both boar and the bull, AQP3 is linked to sperm cryoresistance in boar but not in bulls, while AQP11 is linked to sperm cryotolerance in bull [30–32] but not in boar [28, 33]. Almost all studies suggest that AQP3 and AQP7 [35], as well as AQP11, are linked to spermatozoa cryotolerance. Therefore, a study was conducted to see if ejaculates containing greater levels of AQP3 and AQP7 could adapt to severe osmotic fluctuations via efficient transportation of water and cryoprotective chemical during the cryopreservation process, particularly in the sperm tail, resulting in high motility post-thawing [30]. Interestingly, the same situation occurred with AQP11, and the link between AQP11 and spermatozoa cation channels might help explain why this aquaporin is involved in boar sperm function and cryotolerance [33]. Even though AQP11 has been shown to be permeable to water in liposomes and cultured cells, its permeability to glycerol has not yet been confirmed [36]. In addition, regarding the relationship between these three AQPs and sperm cryoresistance, dissertations have demonstrated that the AQP7 in fresh sperm can predict ejaculate freezability in boar and bull. While AQP3 is

involved in the cryoresistance of boar spermatozoa, AQP11 is only engaged in the cryotolerance of bull spermatozoa. Thus, bull fresh ejaculates containing more AQP11 appear to have better *in vitro* fertilizing capacity after thawing [29, 33].

To summarize, APQ7 is acknowledged as a confirmed biomarker for sperm cryotolerance, however the function and mechanism of action of AQP3 and AQP11 as biomarkers differ greatly between species, and further research is needed to determine their precise function and mechanism of action.

## Cryotolerance biomarkers of seminal plasma

**Antioxidant protection.** Seminal plasma proteins have been linked to semen cryotolerance in various species, suggesting that these proteins could be used as biomarkers. There are several seminal plasma biomarkers of cryotolerance that are associated with antioxidants, as with spermatozoa (Fig 2). Total antioxidant capacity (TAC) could be employed as a biomarker for the selection of sperm cryoresistance in stallion [22]. Another interesting study found that the expression of cell-free seminal mRNAs insulin-like growth factor 1(IGF1) could be used as a positive marker of bull spermatozoa to cryoinjury [37]. According to these workers, IGF1 may play a cryoprotective role in antioxidative protection during semen freezing, thus, the capacity of antioxidants plays an important role in sperm freezing and can predict sperm cryotolerance both in sperm and seminal plasma.

**Stability of sperm plasma membrane.** Because of cryopreservation, sperm plasma membrane becomes destabilized, so there are different proteins having a relationship with the plasma membrane necessitated in the sperm resistance to endure cryopreservation. Horse seminal plasma protein-2 (HSP-2) is a potential seminal plasma biomarker associated with improved semen freezability, although the precise mechanism remains elusive. HSP-2 exhibits similarities with bovine seminal plasma proteins (BSP) A1/A2 [38], known for their efficacy in maintaining sperm membrane integrity during cryopreservation of semen. HSP-2 may offer protective benefits to sperm membranes during freezing procedures for bovine semen, thereby qualifying as a promising biomarker [39].

A study aimed at investigating characteristic proteome variations in boar spermatozoa, and seminal plasma revealed that N-acetyl-β-hexosaminidase (β-HEX) activity in seminal plasma serves as a dependable indicator of boar spermatozoa cryoresistance. β-HEXs represent lysosomal enzymes responsible for hydrolyzing acetylglucosamine and acetylgalactosamine residues from the nonreducing end of oligosaccharide residues in glycoconjugates [40], which has a negative correlation with spermatozoa cryoresistance. The enzyme may bind substantial amounts of zinc ions, which can render to alterations in plasma membrane liquidity and removes zinc ions, while the enzyme from the plasma membrane increases the cryotolerance of boar semen [41].

The contents of Niemann-Pick C2 (NPC2) protein isoforms in seminal plasma can predict ejaculate cryotolerance because the NPC2 has an efficient cholesterol-binding site [42], and the 19 kDa isoform seems to be more able to unite the sperm membrane to protect the spermatozoa by exerting a stabilizing action on the membrane and preventing cholesterol efflux in the process of semen cryopreservation during the cold shock [43]. The NPC2 would impair sperm membrane fluidity and contribute to the creation of lipid-phase separations. As a result, $Ca^{2+}$ inflow will increase, thus membrane integrity could be compromised [44, 45]. Despite this assumption, spermatozoa from NPC2-lacking mice showed increased cholesterol loss from the membrane, indicating that NPC2 is involved in the conservation of cholesterol contents in the sperm membrane in this specie [46].

Similarly, another study, the relationship between three semen plasma proteins (HSP90a, NPC2, and lipocalin-type prostaglandin D synthase-L-PGDS) involved in sperm function revealed that the three proteins tested are linked to sperm freezability in boar and the NPC2 could be a biomarker of sperm freezability in seminal plasma, for that the NPC2 can intervene with the movement of cholesterol from the plasma membrane. The HSP90a participates in sperm capacitation and apoptosis processes, and membrane integrity. The L-PGDS is thought to be a carter of retinoids (retinol and retinoic acid), which restrict the penetrability of the plasma membrane while cooperating with phospholipids, allowing more ions to enter from the outside [46]. This phenomenon may be related to cryocapacitation due to access to ions such as $Ca^{2+}$, which turns out to be an acrosome and hypermotility reaction [47], and the L-PGDS protects sperm from freezing injury through maintaining the integrity of the sperm plasma membrane. The PGDS's participation in modifications of cell membrane permeability during cryopreservation may be explained according to a study [48], which showed that the BSP A1/A2, aSFP, and PGDS could be used as biomarkers for bull semen freezability. For that, the BSP A1/A2 plays a vital function in limiting sperm membrane alterations during cryopreservation, whereas the aSFPs protect sperm membrane preservation against peroxidation.

The nerve growth factor (NGF) was investigated in the study of IGF1 because NGF may play a cryoprotective role in protecting sperm membrane integrity against cryoinjury either alone or in combination with other molecules, and UBE2D3 expression levels were linked to protecting mitochondrial membrane potential during cryopreservation [49]. There are plasma membrane integrity problems and cholesterol efflux during sperm freezing, as well as the irreversible protein aggregation, lateral-phase separation of lipids, and loss of selective permeability. The cryotolerance biomarkers of seminal plasma mainly participate in several aspects of maintaining plasma membrane stability in the procedure of frozen-thaw [50].

The cysteine-rich secretory protein-3 (CRISP-3) genotype has been identified as a biomarker for early selection and detection of stallions with high semen freezability in adulthood [51]. Consequently, extensive investigation has been dedicated to elucidating the precise mechanism of CRISP-3 activity. However, it's worth mentioning that CRISP-3 and CRISP-1 may exhibit overlapping functions in the reproductive process [52]. CRISP-1 acts as a capacitation inhibitor, preventing premature activation of spermatozoa in rats. Meanwhile, CRISP-3 has been proposed as a potential seminal plasma marker in equines with enhanced semen freezability, based on the assessed concentration of this protein in seminal plasma [51]. Notably, CRISP-3 exhibited a higher relative protein content in samples obtained from stallions with GFE. The same research with TAC also found that PUFA could be used as a biomarker for the selection of stallion sperm cryoresistance [22]. Since sperm membranes with higher ratios of sterol to phospholipid are more resistant to a phase transition during cryopreservation, a study of post-thaw semen quality that looked for associations with traditional and novel semen and seminal plasma parameters, suggested that corrected cholesterol could be served as a novel biomarker for predicting human semen post-thaw quality [52]. To sum up, almost all of the research on biomarkers is focused on sperm membrane stability to predict sperm freezing tolerance in seminal plasma.

**Other cryotolerance biomarkers.** The research carried out to detect important biomarkers for prospective freezability of spermatozoa demonstrated that apoptotic markers can be used to predict freezability [53]. Spermatozoa subjected to cryopreservation undergo an apoptotic-like process as cryopreservation stimulates this process, resulting in cell death. Moreover, apoptosis-like phenomena are stimulated during cryopreservation, particularly in the stallion with a high percentage of spermatozoa depicting heterogeneous mitochondria with high and low mitochondrial membrane potential within the same sperm cell. Mitochondria are the most vulnerable subcellular structure to cryoinjury [54]. This supports more

investigation into this organelle for the improvement of its value as a sperm cryopreservation forecast [53].

Research about the sensitivity of dog sperm cells for extracellular $Ca^{2+}$/$Ca^{2+}$-ionophore challenge as compared to the detrimental effects of an optimized thawing protocol demonstrated that $Ca^{2+}$-ionophore treatment, followed by simultaneous determination of fluorescein-conjugated peanut agglutinin (PNA-FITC) and ethidium homodimer 1 (EthD-1) staining can be used to predict the cryopreservability of ejaculates from dogs [55]. It implies that at least a part of the acrosomal deterioration imposed by cryopreservation is induced via a similar $Ca^{2+}$ entry as induced by the $Ca^{2+}$-ionophore treatment in the apical part of the sperm head, while the cryopreservation-induced perturbations of the plasma membrane structure have been reported to alter the $Ca^{2+}$ transport over the sperm plasma membrane, resulting in extremely high $Ca^{2+}$ influx, consequently, the deterioration rates resulted from $Ca^{2+}$-ionophore treatment are good tools to predict the sperm motility characteristics of boar semen after cryopreservation [56].

It is documented that miRNAs found in spermatozoa and seminal plasma small extracellular vesicles (EVs) can be used as indicators for boar semen cryoresistance [57]. Interestingly, these miRNAs augmented in ejaculates with low freezability, probably indicating that miRNAs present in EVs from seminal plasma and in sperm cells could be tangled in depressed sperm cryotolerance, possibly triggering detrimental energy production pathways or alternating sperm development. Modulated biological pathways by miR-130a and miR-9 reported augmented in EVs from the seminal plasma of PFE, the most destructively controlled path of fatty acid biosynthesis, which is submerged in various biological and metabolic processes including energy source in spermatozoa and cell membrane scaffolding. Furthermore, fatty acid molecules are imperative for generating energy via oxidative phosphorylation, and glycolysis which is thus supporting sperm motility, and most essentially, upholding the steadiness and role of the plasma membrane [58]. As a result, fatty acid disfunction may reduce plasma membrane integrity and sperm motility leading to poor survival of spermatozoa, thus boar sperm cells with low viability are more likely to be cryodamaged and become dead. In summary, swine spermatozoa miRNAs and seminal plasma EVs in ejaculates could be used as indicators for spermatic cryoresistance [57].

Serum levels of Se, Zn, and Fe are linked to nearly all characteristics of frozen-thawed sperm. Se is tangled in spermatogenesis and is a crucial element of glutathione peroxidase 4 (GPX4). Concentration of Se in blood is necessary for normal spermatogenesis, while Zn is essentially required in the maturation of epididymal sperm and capacitation. Whereas morphology, viability, progressive motility, and tail membrane integrity are linked with seminal plasma Fe levels. As a result, these three elements, i.e., Se, Zn, and Fe concentrations in blood serum and Zn and Fe in seminal plasma of bull can be used as freezability biomarkers [59].

## Discussion

### Biological functions of biomarkers

The sperm membrane is involved in the material exchange, metabolism, information exchange, and other functions of sperm cells, while the structural and functional integrity of the membrane is critical to the physiological functions of the entire sperm. From the above discussion it can be extracted that one of the first structure concerned by cryopreservation is the sperm plasma membrane [59], which contains low levels of cholesterol and high levels of unsaturated phospholipids. Redistribution of phospholipids through the membrane occurs and some of these are converted from fluid to gel state earlier than others due to structural variations, ensuing in a lipid phase separation during freezing [60]. Consequently, the lipid-

protein interactions required for proper membrane activity are disturbed [61], and some sperm surface proteins, as well as membrane proteins, are lost or translocated with the consequent loss of their function. Substances involved in oxidative stress, protein phosphorylation, lipid metabolism, and glucose metabolism can affect the stability of sperm cell membrane and could also be the biomarkers to predict semen cryotolerance.

Cryopreservation also stimulates substantial alterations in the abundance or distribution of ROS scavengers. Related antioxidant enzymes such as superoxide dismutase (SOD), glutathione reductase (GR), and glutathione peroxidase are redeployed on the surface of ram sperm. Moreover, the abridged SOD antioxidant activity and decreased glutathione (GSH) in sperm after cryopreservation, could expound partially that escalated vulnerability of frozen-thawed sperm to undergo oxidative damage and lipid peroxidation [62].

The energy required to maintain sperm motility is produced by two main metabolic pathways: glycolysis and oxidative phosphorylation [63]. In ram sperm, comparative proteomics investigations implicated oxidative phosphorylation and glycolysis [64]. Among them, the glucose-6-phosphate isomerase (GPI) and triose phosphate isomerase (TPI) could be biomarkers. Reasons for variations in this proteomics are yet to be resolved. Some researchers purported a few mechanisms to clarify such variations that could involve oxidation, protein degradation, and translocation phosphorylation of tyrosine [65, 66].

Cryopreservation of sperm has a detrimental effect on cytoskeletal proteins. This helps to explain why several cytoskeletal proteins in spermatozoa decrease in number or change in location after freezing and thawing [67]. Because cytoskeletal proteins are implicated in integrity of axoneme preservation, these findings have substantial implications for sperm movement. TEKT4, a cytoskeletal protein, could be used as a diagnostic for sperm cryoresistance.

Another protein, AQP plays a critical role in the perviousness of plasma membrane to water and cryoprotectants through cryopreservation. The transport rates of water and cryoprotectants are much higher in facilitated diffusion where AQPs are involved. Therefore, the GLPs are good sperm freezability biomarkers.

## Conclusions and future perspective

This review summarizes the cryotolerance biomarkers of spermatozoa and seminal plasma and other biomarkers, as well as the molecular mechanisms of biomarkers involved in the frozen-thawed procedure of spermatozoa. The study of biomarkers of ejaculates with high and low freezability can be significant for the livestock industry, as well as the administration of germplasm banks, thus these banks can select the high freezability ejaculates before the cryopreservation process and produce excellent grade cryopreserved sperm on a commercial scale. On the other side, the study of sperm cryotolerance biomarkers and molecular mechanisms which can promote the study of PFE to achieve better sperm freezing, promote the research of better semen diluents and semen freezing procedures, and make important contributions to animal husbandry and endangered species as well as the biodiversity conservation.

## Supporting information

**S1 Checklist. PRISMA 2020 checklist.**
(DOCX)

## Author Contributions

**Conceptualization:** Jiajia Suo, Qinghua Gao.

**Data curation:** Jiajia Suo, Yanling Zheng, Fayi Xiao, Ruchun Li, Fei Huang.

**Formal analysis:** Jiajia Suo, Peng Niu, Wei Zhu, Xiaoxia Du.

**Software:** Jianxiu He.

**Supervision:** Qinghua Gao.

**Validation:** Fei Huang, Qinghua Gao.

**Visualization:** Jianxiu He.

**Writing – original draft:** Jiajia Suo, Jieru Wang, Qinghua Gao, Ahrar Khan.

**Writing – review & editing:** Jieru Wang, Qinghua Gao, Ahrar Khan.

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
