## [Decision Letter · Decision Letter 0]

20 Feb 2024

PONE-D-23-40697RECENT ADVANCES IN CRYOTOLERANCE BIOMARKERS FOR SEMEN PRESERVATION IN FROZEN FORMPLOS ONE

Dear Dr. Khan,

Thank you for submitting your manuscript to PLOS ONE. After careful consideration, we feel that it has merit but does not fully meet PLOS ONE’s publication criteria as it currently stands. Therefore, we invite you to submit a revised version of the manuscript that addresses the points raised during the review process.

Please submit your revised manuscript by Apr 05 2024 11:59PM. Please include the following items when submitting your revised manuscript:A rebuttal letter that responds to each point raised by the academic editor and reviewer(s). You should upload this letter as a separate file labeled 'Response to Reviewers'.A marked-up copy of your manuscript that highlights changes made to the original version. You should upload this as a separate file labeled 'Revised Manuscript with Track Changes'.An unmarked version of your revised paper without tracked changes. You should upload this as a separate file labeled 'Manuscript'.

We look forward to receiving your revised manuscript.

Kind regards,

Joël R Drevet, Ph.D.

Academic Editor

PLOS ONE

2. Please identify your study as ""systematic review"" in the title of your manuscript.

4. We note that Figure 3 in your submission contain copyrighted images. All PLOS content is published under the Creative Commons Attribution License (CC BY 4.0), which means that the manuscript, images, and Supporting Information files will be freely available online, and any third party is permitted to access, download, copy, distribute, and use these materials in any way, even commercially, with proper attribution. For more information, see our copyright guidelines: http://journals.plos.org/plosone/s/licenses-and-copyright.

a. You may seek permission from the original copyright holder of Figure 3 to publish the content specifically under the CC BY 4.0 license. 

Additional Editor Comments:

TTwo experts in the field have evaluated your submission. They consider it worthy of publication, but both request a few modifications. Although they have both qualified the requested changes as minor, I urge the authors to follow their requests scrupulously, particularly with regard to missing data and references, and parts of the text that are too similar to previously published contributions. For these reasons, my final assessment is "major revision".

Reviewers' comments:

Reviewer's Responses to Questions

**Comments to the Author**

1. Is the manuscript technically sound, and do the data support the conclusions?

Reviewer #1: Yes

Reviewer #2: Yes

2. Has the statistical analysis been performed appropriately and rigorously? 

Reviewer #1: N/A

Reviewer #2: N/A

3. Have the authors made all data underlying the findings in their manuscript fully available?

Reviewer #1: No

Reviewer #2: Yes

4. Is the manuscript presented in an intelligible fashion and written in standard English?

Reviewer #1: Yes

Reviewer #2: Yes

5. Review Comments to the Author

Reviewer #1: This literature review explores a compelling subject related to semen cryotolerance biomarkers, offering a substantial amount of information. Upon examination, a notable gap in the existing body of research becomes apparent, particularly in the examination of metabolites previously identified as biomarkers. Notably absent are contributions from the referenced research group, and the absence of an opinion based on the research team's experience is evident, presenting an opportunity to enhance the text through the inclusion of such insights. This additional layer of analysis could significantly enrich the overall depth and scholarly value of the review.

Reviewer #2: This manuscript represents a comprehensive overview of studies that have addressed a topic that is highly relevant to the field of animal andrology - molecular markers of cryotolerance or resistance in semen.

The search strategy and data evaluation is clearly presented, and the collected findings are well-presented.

I do have several comments that could be considered by the authors:

- Please, modify the section on the search strategy (lines 84-90). It is not necessary to repeat that the search/analysis included paper published until March 2023.

- Several sections of the manuscript (specifically paragraphs regarding the fatty acids, aquaporins, HSP-2, CRISP-3, beta-HEX) resemble the review published by Duracka et al. (https://doi.org/10.3390/ijms24043379). Please, add or modify these sections to distinguish your information from the above-mentioned paper.

- Please, add the animal species each referenced study was focused on, where missing.

- The section on improvements to PFE cryopreservation is very vague. There are no references, and overall, this section does not provide much insight. Either delete it or it should be extended to include different strategies, either physico- chemical or much more additives that could be used (not just extracts).

- Aquaporins seem to hold a specific position in the manuscript. Technically, they are sperm markers, yet is there a reason as to why they deserve a separate section?

- It would be of benefit, if within the seminal plasma/sperm/other markers of cryobehavior, the authors would create sub-sections, with individual markers being categorized – such as markers related to antioxidant protection, stress response, transport, metabolism, etc.

6. PLOS authors have the option to publish the peer review history of their article (what does this mean?). If published, this will include your full peer review and any attached files.

Reviewer #1: No

Reviewer #2: No

---

## [Author Response · Author response to Decision Letter 0]

8 Apr 2024

We have addressed all points raised by the Editor/Reviewers and our responses to each point is presented in rebuttal-report. All changes in the revised version are in red colored font.

---

## [Decision Letter · Decision Letter 1]

29 Apr 2024

RECENT ADVANCES IN CRYOTOLERANCE BIOMARKERS FOR SEMEN PRESERVATION IN FROZEN FORM - A SYSTEMATIC REVIEW

PONE-D-23-40697R1

Dear Dr. Ahrar Khan,

We’re pleased to inform you that your manuscript has been judged scientifically suitable for publication and will be formally accepted for publication once it meets all outstanding technical requirements.

Kind regards,

Joël R Drevet, Ph.D.

Academic Editor

PLOS ONE

**Comments to the Author**

1. If the authors have adequately addressed your comments raised in a previous round of review and you feel that this manuscript is now acceptable for publication, you may indicate that here to bypass the “Comments to the Author” section, enter your conflict of interest statement in the “Confidential to Editor” section, and submit your "Accept" recommendation.

Reviewer #1: All comments have been addressed

Reviewer #2: All comments have been addressed

2. Is the manuscript technically sound, and do the data support the conclusions?

Reviewer #1: (No Response)

Reviewer #2: Yes

3. Has the statistical analysis been performed appropriately and rigorously? 

Reviewer #1: (No Response)

Reviewer #2: Yes

4. Have the authors made all data underlying the findings in their manuscript fully available?

Reviewer #1: (No Response)

Reviewer #2: Yes

5. Is the manuscript presented in an intelligible fashion and written in standard English?

Reviewer #1: (No Response)

Reviewer #2: Yes

6. Review Comments to the Author

Reviewer #1: (No Response)

Reviewer #2: The paper has undergone a solid revision and all the comments raised by the reviewers have been addressed accordingly. I have no further questions.

7. PLOS authors have the option to publish the peer review history of their article (what does this mean?). If published, this will include your full peer review and any attached files.

Reviewer #1: No

Reviewer #2: No

---

## [Editor Report · Acceptance letter]

9 May 2024

PONE-D-23-40697R1 

PLOS ONE

Dear Dr. Khan, 

I'm pleased to inform you that your manuscript has been deemed suitable for publication in PLOS ONE. Congratulations! Your manuscript is now being handed over to our production team.

Kind regards, 

on behalf of

Prof. Joël R Drevet 

Academic Editor

PLOS ONE